

# Density of intertidal barnacles along their full elevational range of distribution conforms to the abundant-centre hypothesis

Ricardo A. Scrosati and Matthew J. Freeman

Department of Biology, St. Francis Xavier University, Antigonish, NS, Canada

## ABSTRACT

The abundant-centre hypothesis (ACH) predicts that the density of a species should peak at its distribution centre and decrease similarly towards distribution margins. The ACH has been deduced from a theory that postulates that environmental conditions should be most favourable for a species at the centre of its distribution. This idealised density pattern, however, has been supported by limited field studies, as natural patterns are often more complex. It is thus of interest to examine under what conditions compliance with the ACH could be favoured. Such conditions could be smooth environmental gradients with limited habitat patchiness throughout the distribution range of a species. Thus, we tested the ACH by measuring the density of an intertidal barnacle (*Semibalanus balanoides*) across its full vertical distribution range (from low to high elevations) on a rocky shore with similar substrate properties across elevations. To do a reliable test, we surveyed eight elevation zones applying an equal sampling effort in each zone. Average barnacle density conformed to the ACH, as it peaked at the middle of the vertical distribution range of this species. The same underlying theory predicts a similar unimodal pattern for maximum body size, but this trait was decoupled from density, as maximum barnacle size increased from low to high elevations. Overall, although the ACH is not a universal predictive tool as once envisioned, it may predict some cases well, as shown by this study. Therefore, the ACH should not be discarded completely, and its domain of application should be further evaluated.

## INTRODUCTION

The abundant-centre hypothesis (ACH) predicts that the density of a species should peak at its distribution centre and decrease similarly towards the distribution margins. The ACH has been deduced from a theory that postulates that environmental conditions should be most favourable for a species at its distribution centre (*Brown, 1984*). These concepts were conceived without particularly considering the extent of species distributions, which can be large or small depending mainly on limits imposed by the physical environment. In addition, distribution ranges can be examined in various

Corresponding author
Ricardo A. Scrosati, rscrosat@stfx.ca

directions (e.g., horizontally along forest transects or vertically across mountain elevations). The convenient simplicity of the ACH has been used to generate more complex hypotheses predicting, for example, changes in population persistence, genetic diversity, speciation potential, and consumer control from the distribution margins of a species to its distribution centre (*Lawton et al., 1994*; *Lesica & Allendorf, 1995*; *Channell & Lomolino, 2000*; *Holt & Keitt, 2000*; *Eckert, Samis & Lougheed, 2008*; *Dixon, Herlihy & Busch, 2013*; *Micheletti & Storfer, 2015*). However, field tests of the ACH have produced mixed results. While some surveys have supported the ACH, others have failed to do so, finding various levels of departure from the idealised unimodal trend for density (*Sagarin & Gaines, 2002*; *Rivadeneira et al., 2010*; *Tam & Scrosati, 2011*; *Pironon et al., 2017*; *Dallas, Decker & Hastings, 2017*). Such departures often result, in part, from the patchiness that abiotic conditions generally display across space in the real world (*Helmuth et al., 2002*; *Gilman, 2005*; *Tam & Scrosati, 2011*; *Martínez-Meyer et al., 2013*).

While the ACH is not a universal predictive tool as once envisioned, it could retain some usefulness in certain cases. One such case might be species distribution ranges that span smooth environmental gradients. In rocky intertidal habitats, for example, there is a strong change in abiotic conditions from low to high elevations because of tides (*Raffaelli & Hawkins, 1999*; *Menge & Branch, 2001*). While low elevations remain submerged most of the time, high elevations are frequently exposed to the air. Thus, for exclusively intertidal species, which are adapted to some degree of aerial exposure, the upper and lower margins of the intertidal range represent environmental extremes. Therefore, unless substrate properties are too patchy across elevations, sessile species that are only intertidal and span the full intertidal range could experience the most favourable conditions at middle elevations. This line of reasoning suggests that the ACH could hold for such species. On Atlantic Canadian rocky shores, the barnacle *Semibalanus balanoides* is an exclusively intertidal sessile species whose vertical distribution range spans low to high elevations (*Scrosati & Heaven, 2008*). Thus, under the above considerations, it is a suitable species for ACH testing.

Field tests of the ACH are most reliable when surveys cover the full distribution range of a species and apply the same sampling effort across the range (*Gaston, 2003*). However, due to factors that limit site access (e.g., rough topography or poor roads), these two key requirements have not always been met (*Dallas, Decker & Hastings, 2017*). Such difficulties generally do not apply to intertidal studies, as the full vertical distribution of intertidal species can be surveyed easily during low tides taking data at several elevation zones. Thus, some studies counted intertidal organisms across several elevation zones (*Bell, 1979*; *Brown, 1984*), but replication was either poor or not reported and statistical analyses not done. In 2011, a well-replicated study tested the ACH for *S. balanoides* using density data for eight elevation zones spanning the full vertical distribution range of this species. Although statistical analysis revealed a unimodal density trend across elevations, density peaked at a higher elevation than expected, thus providing weak support for the ACH (*Scrosati, Grant & Brewster, 2012*). In ecology, equally important as replication of sampling units is the replication of whole studies. Repeating studies in the future allows for the determination of pattern consistency, which is essential in the

process of developing and refining theories (*Krebs, 1999*; *Ford, 2000*). Therefore, 7 years after the study that tested the ACH using *S. balanoides* density data (*Scrosati, Grant & Brewster, 2012*), we replicated the survey with the objective of evaluating the temporal consistency of the pattern.

Body size is an important emergent property of organisms (*Lika, Augustine & Kooijman, 2019*). It is related to ecological performance (*Brown et al., 2004*) and affects upper trophic levels through consumptive interactions (*Dunkin & Hughes, 1984*; *Carroll & Wethey, 1990*). The theory from which the ACH has been deduced (conditions should be best for a species at its distribution centre; *Brown, 1984*) can also be employed to predict spatial trends in maximum body size. Specifically, maximum body size can be predicted to peak at the distribution centre of a species and decrease towards its distribution margins (*Scrosati, Grant & Brewster, 2012*). Some studies have measured how the body size of invertebrates changes along vertical intertidal gradients (*Vermeij, 1972*; *Bertness, 1977*; *McCormack, 1982*; *McQuaid, 1982*; *Hobday, 1995*), but those studies only considered mobile invertebrates. Data measured at a single point in time describing body size of mobile species across intertidal elevations may be highly influenced by recent abiotic conditions, as mobile species move up and down the shore when conditions change (e.g., hot vs cool days). In contrast, body size data for sessile species (such as barnacles) integrate much better the environmental influences over the long term. The study done in 2011 on *S. balanoides* also tested the above prediction for maximum body size and found that this trait varied unimodally across elevations but, like density, peaking at a higher elevation than expected (*Scrosati, Grant & Brewster, 2012*). Therefore, the present study replicates that study also with the goal of evaluating pattern consistency.

## MATERIALS AND METHODS

We did this study at Tor Bay Provincial Park (45°11′N, 61°21′W), on the Atlantic coast of Nova Scotia, Canada, on 25 November 2018. We surveyed wave-sheltered intertidal habitats, which are protected from direct oceanic swell by outer islets and are characterised by values of daily maximum water velocity of 3 to 6 m s$^{-1}$ (*Scrosati & Heaven, 2007*). The surveyed habitats are composed of stable bedrock with a similar rugosity and a steep slope from low to high elevations. On this coast, *S. balanoides* (Fig. 1) is the only intertidal barnacle species (*Scrosati & Heaven, 2007*). Organisms of this species can live for a few years (*Anderson, 1994*). The vertical intertidal range is 1.8 m on this coast. To test our hypotheses, we surveyed intertidal habitats spanning several meters along the coastline. In these habitats, we first used tide tables to divide the intertidal range into nine elevation zones of equal vertical extent (0.2 m). As barnacles were absent at the lowest zone (between 0 and 0.2 m of elevation), the full vertical distribution range of this barnacle was deemed to span 1.6 m (the eight zones between 0.2 and 1.8 m of elevation, relative to chart datum). For each of those zones, we determined the density and maximum body size of *S. balanoides* for 10 quadrats (20 × 20 cm) randomly established following the coastline. We calculated density as the number of barnacles found in a quadrat divided by quadrat area (4 dm$^2$). We measured maximum body size as the basal diameter (to the nearest 1 mm) of the widest barnacle found in a quadrat using a ruler.

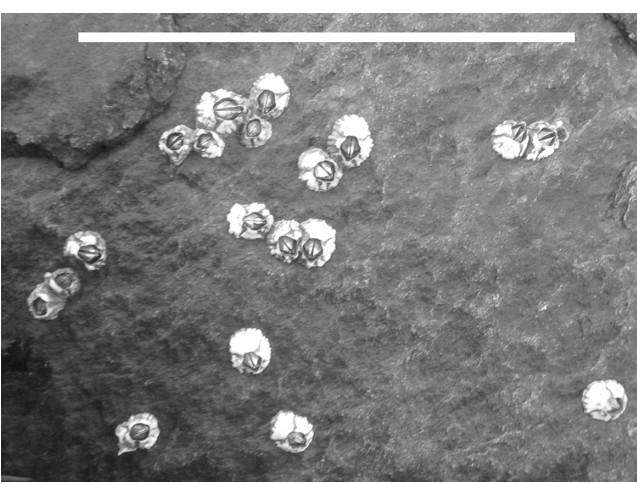

**Figure 1 Intertidal barnacles (*Semibalanus balanoides*) from the studied coast.** Scale bar: 10 cm. Photograph taken by R. A. Scrosati.

We determined how barnacle density and maximum body size changed across elevations by calculating a generalised additive model (GAM) with a Gamma distribution (*Zuur et al., 2009*) for each of these two barnacle traits, elevation zone being the independent variable in both models. A GAM identifies the most suitable functional relationship without any pre-set shape in mind. For both cases, the basis dimension ($k$) for the smoother was 5 and predictor-vs-residuals plots did not show any evident structure. To evaluate the level of support for each GAM, we compared its AICc value (corrected Akaike Information Criterion) with the AICc value of the corresponding intercept-only model (the model for the same barnacle trait including an intercept but not elevation zone). A GAM was deemed to have substantial support if its AICc score was lower than the AICc score of the intercept-only model by more than six units of difference (*Harrison et al., 2018*). For each barnacle trait, we calculated the evidence ratio to determine specifically how many times more plausible the GAM was relative to the intercept-only model (*Anderson, 2008*). Finally, for each GAM, we determined the percentage of variation in either density or maximum body size explained by elevation by calculating the explained deviance (*Zuur et al., 2009*). We conducted these analyses with R version 3.5.1 (*R Development Core Team, 2018*), using the mgcv package to calculate the four models and the MuMIn package to calculate the corresponding AICc scores. The full data set used for this paper is available from the figshare online repository (*Scrosati & Freeman, 2019*).

## RESULTS

The density of *S. balanoides* followed a unimodal trend across the vertical distribution range of this species, peaking at the centre of this range (Fig. 2). The GAM describing this relationship had an AICc score of 458.8, more than six units lower than the AICc score of 472.2 for the intercept-only model. Based on the evidence ratio, the GAM was 816 times more plausible than the intercept-only model. The estimated degrees of freedom

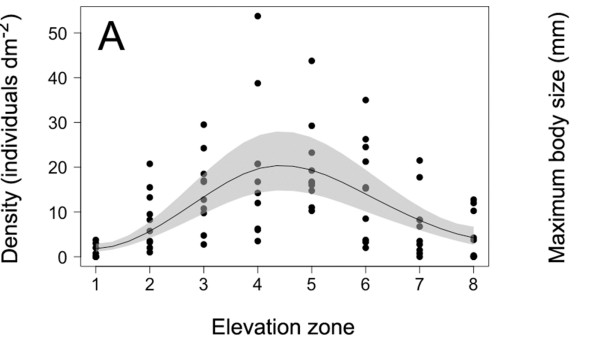 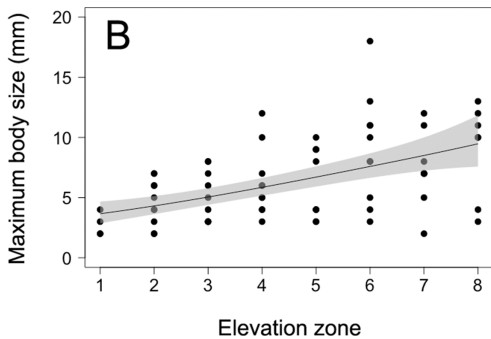

**Figure 2 Barnacle density and maximum body size.** (A) Density and (B) maximum body size (maximum basal diameter) of barnacles (*Semibalanus balanoides*) at the eight intertidal elevation zones (zone 1 being the lowest) that span the full vertical distribution range of this species. In each graph, the solid line is the GAM depicting the functional relationship and the shaded area indicates the 95% confidence band.

for the GAM were 2.5. The percentage of variation in barnacle density explained by intertidal elevation (explained deviance) was 16%.

The maximum body size of *S. balanoides* increased relatively consistently from the lower distribution margin of this species to its upper distribution margin (Fig. 2). The GAM describing this trend had an AICc score of 339.8, more than six units lower than the AICc score of 359.2 for the intercept-only model. Based on the evidence ratio, the GAM was 16,564 times more plausible than the intercept-only model. The estimated degrees of freedom for the GAM were 1.4. The percentage of variation in maximum body size explained by intertidal elevation was 27%.

## DISCUSSION

The 2018 data on *S. balanoides* supported the ACH. Density exhibited a unimodal pattern across the full vertical distribution range of this species and peaked at the middle of the range. Intertidal elevation explained only a moderate amount of the observed variation in density, but the overall trend predicted by the ACH was nonetheless highly supported, based on the evidence ratio. The 2011 study, done on the same coast with the same field methodology, had also found a unimodal pattern for *S. balanoides* density across elevations, but density then peaked at a higher elevation than predicted by the ACH (*Scrosati, Grant & Brewster, 2012*). Thus, while both studies suggest that a unimodal vertical distribution pattern is common for this species, density may or may not peak where the ACH predicts. Nonetheless, the 2018 study reveals that the idealised abundant-centre distribution can indeed occur. Therefore, although now known not to be a universally valid concept, the ACH does retain some usefulness. This example is thus a valuable addition to the limited number of studies that have supported the ACH through surveys on a variety of species (*Sagarin & Gaines, 2002*; *Tuya, Wernberg & Thomsen, 2008*; *Rivadeneira et al., 2010*; *Pironon et al., 2017*).

Explaining what factors modified (2011 vs 2018) the elevation with the highest density of *S. balanoides* is difficult given the absence of experimental data for the studied coast. Future efforts aiming to explain interannual differences could examine interannual

changes in abiotic factors (*Jones, Southward & Wethey, 2012*), positive and negative effects of algal canopies (*Beermann et al., 2013*), and consumptive (*Menge, 1976*) and nonconsumptive (*Ellrich, Scrosati & Molis, 2015*) effects of predators. Although barnacles compete with mussels on other rocky shores (*Lubchenco & Menge, 1978*; *Peterson, 1979*), mussels are almost absent in the studied sheltered habitats (*Scrosati & Heaven, 2007*).

The 2018 data on the maximum body size of *S. balanoides* failed to support the hypothesised trend deduced from the theory that also underlies the ACH. In 2011, maximum body size had followed (like density) a unimodal pattern across elevations, peaking at a higher elevation than the distribution centre (*Scrosati, Grant & Brewster, 2012*). However, in 2018, maximum body size increased relatively smoothly from low to high elevations. Thus, these two studies suggest that, for barnacles, elevational patterns in maximum body size are less stable over the years than those of density. Patterns in maximum body size may (2011) or may not (2018) be coupled to density patterns. For *S. balanoides*, no density-dependence should have influenced the relationship between density and maximum body size across elevations, because crowding did not occur on the studied shore in any of the two studied years (Fig. 1; *Scrosati, Grant & Brewster, 2012*). Among invertebrates, there are other examples for which body size increases relatively consistently towards one of the distribution margins (*Rivadeneira et al., 2010*).

## CONCLUSIONS

Overall, our research with *S. balanoides* emphasises the notion that, while the ACH is an oversimplistic concept that cannot predict all species distributions, it can predict some cases relatively well. Thus, the ACH should not be discarded completely, and its domain of application should be further evaluated.

## ACKNOWLEDGEMENTS

We thank two anonymous reviewers, Donald Kramer, Derrick Lee, and Martin Lavoie for their constructive comments on an earlier version of this paper.

### Funding

This study was funded by a Discovery Grant (#311624) awarded by the Natural Sciences and Engineering Research Council (NSERC) to Ricardo A. Scrosati. The funders had no role in study design, data collection, and analysis, decision to publish, or preparation of the manuscript.

### Grant Disclosure

The following grant information was disclosed by the authors:
Natural Sciences and Engineering Research Council: Discovery grant (#311624).

### Competing Interests

The authors declare that they have no competing interests.

## Author Contributions

- Ricardo A. Scrosati conceived and designed the experiments, performed the experiments, analysed the data, contributed reagents/materials/analysis tools, prepared figures and/or tables, authored or reviewed drafts of the paper, approved the final draft.
- Matthew J. Freeman performed the experiments, authored or reviewed drafts of the paper, approved the final draft.

## Data Availability

Scrosati, Ricardo A.; Freeman, Matthew J. (2019): Data on barnacle density and maximum body size from Nova Scotia (2018). figshare. Dataset. DOI 10.6084/m9.figshare.7547825.v1.

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
