# Peer review of "Density of intertidal barnacles along their full elevational range of distribution conforms to the abundant-centre hypothesis"

_PeerJ, doi:10.7717/peerj.6719_

## Round 0.1 · original submission · Major Revisions

Overview
This study examined the vertical distribution of density and maximum size of barnacles in the intertidal zone of a steep shoreline to test the Abundant Centre Hypothesis. Density peaked at mid-height, providing support for the hypothesis, but maximum size peaked at the highest zone, contrary to prediction.

Both reviewers, who have experience with this theory, indicated important concerns. These include the application of a biogeographical model to a local within-population habitat gradient, the relevance of maximum size to the theory, and the statistical analysis. In addition, they provided many useful comments that could improve the presentation. I have some concerns related to those raised by the reviewers and some of my own.

Overall, the manuscript is very well written and I have very few suggestions for grammatical or word choice changes.

Editor's comments
Major concerns
1) It is not clear how the local scale of the test is related to the larger scale of the hypothesis. Although not familiar with the hypothesis myself, when I first read the manuscript, I assumed that the prediction was related to density in the center of the geographical range. Thus, I was surprised that it could be tested within a local population on an environmental gradient. On re-reading, I note that the manuscript used the word 'distribution' and never explicitly mentioned geographical range. However, the examples of implications (L34) would all relate to geographical range and not to a local gradient. Even if the hypothesis was originally formulated to apply to a wide range of spatial scales, the application to these different scales must surely differ as a result of difference in the underlying processes. In agreement with the reviewer, I think that the Introduction must clearly address the relevance of the spatial scale of the test. If the hypothesis cannot be clearly applied at this scale, the entire manuscript will need to be rewritten.
2) I also found the body size prediction ambiguous. The manuscript indicates that maximum body size can be predicted to peak at the center of the distribution (L82), but there is no reference provided (with similar issues on L 166). Does this indicate that this is an original prediction in the present manuscript? If so, the logic needs to be more fully developed and the writing needs to clearly indicate that this is a hypothesis that the authors of this manuscript are putting forward. If the logic has been developed elsewhere, citations are needed. A reviewer also raises concerns about this hypothesis.
3) The manuscript uses the expression 'remarkable fit' in the title, abstract and elsewhere in the manuscript (L144), but does not justify what is so remarkable about it. Is it just that other studies including a previous study of this species and location have not found this pattern, that the peak density is very precisely at the center (this does not have statistical support and may not be correct), that the explained variance is very high (not supported statistically), that the effect size is very large (not discussed), or something else? If it is important to retain the concept of a striking pattern, there must be a logical discussion to support it.

Other suggestions
L2. The second part of the title ' . . . and decoupling from body size data' is not at all clear without reading the article, does not seem to be grammatically correct, and the relevant pattern is surely body size and not your data.
L22. I also have doubts about the expression 'decoupled from density' because there does not seem to be a necessary logical coupling. Also, L173.
L30. You need to cite the originator of the hypothesis at first description. I infer that this may be Brown 1984, but the citation in the second sentence implies that his article could be a review clarifying the origin of the hypothesis.
L33. You probably mean hypotheses (plural)
L38 (and elsewhere). It would be useful to separate the studies supporting and not supporting the hypothesis in this sentence. A reviewer notes that there may be excessive numbers of citations. If there is a reason to have a complete list, multiple citations could be justified. However, if your point is simply that some studies support the hypothesis and other do not, citing a recent manuscript with a reasonably thorough review of the literature may be sufficient. The relevance of repeating the long list of references on L176-179 is not clear because the statement is quite vague.
L66. I do not agree with the reviewer that 'well-replicated' is a value judgement on your own previous study because you are addressing the issue of replication here. However, you could change the sentence by indicating the number of replicates instead.
L71. While I agree that replicating experiments is important in ecology, the studies of concern here are not true experiments but observational studies. To make the statement relevant, consider switching 'experiments' to 'studies'.
L72. I don't know what you mean by repeating studies in the future.
L86. Is 'one-time data' a recognized term in ecology? It seems a bit like slang, and I don't know what it really means.
L89. I think 'in contrast' would better express your meaning than 'on the contrary'.
L102. As a reviewer suggests, you might insert a sentence or two about the ecology and life history and its geographical range for readers who are not familiar with marine organisms. However, a map of the range, as the reviewer suggests, would be excessive.
L108. Explain in a bit more detail how you randomly assigned the sampling location. A reviewer refers to transects, but your description seems not to involve transects. Some clarification of the extent of the study area and the randomization procedure might help. Also, is it relevant to determine how you measured the vertical position along the shore?
L110. How was the size measured?
L132. Scientific notation would be better, I think.
L144. Insert 'only' after 'elevation'.
L148. Please indicate where the peak was. Ideally, you would have confidence intervals around the peaks and indicate whether the two peaks differed significantly.
L153. It seems to me that 'valuable' is a value judgement that you cannot really make at this stage. Why not just state that is an addition. Do you mean to imply that these 5 studies are the only ones that support the hypothesis? How many have failed to support it? Consider whether this distinction is better addressed in the Introduction or here in the Discussion. Is there any point in returning to the issue of scale and indicate how the supporting and non-supporting tests break down in terms of spatial scale of the test system? Perhaps also specify other potentially relevant issues such as marine vs. terrestrial, mobile vs. sedentary. This might lead to a discussion of why the pattern holds in this system. Do these considerations add to the significance of your contribution, e.g. first to show support in a sedentary marine species?
Fig. 2. In the caption specify the body size measure used. To maintain the relative independence of the figures from the text, please clarify the meaning of the elevation zones indicating precisely how they relate to vertical tidal positions.

Reviewer 1 ·

Basic reporting

Comments on the manuscript peerJ - #34082 “Remarkable fit of barnacle density data to the abundant-centre hypothesis and decoupling from body size data”.


1. Basic Reporting

This article presents a test of the “Abundant Center Hypothesis” formulated by Brown in 1984 on a population of barnacles, focusing on the vertical distribution of the species and its upper and lower distribution limit. Authors have sampled a total 9 transects all located in the same bay in Canada. They analyzed how density and body size vary according to depth.

Regarding the form of this work, I’d say that the writing is good. Yet I think that the way authors cite literature references is quite odd, using 8 to 10 references at a time, which does not help to reader to understand which reference supports the point presented. Regarding the quality of the bibliography, while the biogeographic aspect seems appropriate, I feel that the scientific background on barnacle ecology could be weak (see my comment below). Data are shared through a figshare file with a DOI, and figures are of sufficient quality to be published.

I acknowledge that repeating experiment is a crucial step in ecology, and that this should be valorized and encouraged.

Experimental design

2. Experimental design

The research aims at testing the ACH, which has originally been conceived to explain biogeographical patterns. While I understand that such pattern can be detect at different spatial scale, authors only consider one population, which is not enough to discuss an ACH pattern. First, a single population cannot be representative of the general distribution pattern of a species, even on a very local scale when testing for vertical distribution. Addition of some other populations (even locally) would have helped to contextualize this result. Second, if authors want to discuss the ACH at a local scale (which is an interesting approach) they need to make it more explicit and add relevant literature on micro ecological variation and its impact on population characteristics. In particular, some authors have modified the ACH to formulated a ‘niche centre’ hypothesis in which traits (such as abundance) peak at the niche centre instead of the geographic centre (which can match or not the niche centre). Such literature could help to strengthen the reasoning.

Validity of the findings

3. Validity of the findings

As stated above, the finding do not bring insightful result to the biogeographical hypothesis tested, mainly due to the very limited sampling effort. Results could be more informative if presented in a context of population ecology instead of biogeography.

Additionally, despite I’m not a statistician, I think that the method to estimate body size trend is flawed. Actually, authors sampled the biggest animal of each quadrat to measure its size and estimate a trend across vertical distribution. This is wrong as extremes are not representatives of the mean of the population, and they are dependent of the variance of the trait considered. If, across a vertical distribution gradient, the mean body size of individuals remains constant BUT the variance increases, sampling the biggest individuals would ‘artificially’ lead authors to detect an increase in body size.
Such problem could be solved following two approaches :
- randomly sampling few individuals within each quadrat, and re-calculate the trend
- re-writing the manuscript top make this point clear and state that this manuscript focuses on maximum size and is not representative of the global trend of barnacle size across vertical distribution.

Additional comments

4. General comments

I have some additional concerns regarding the way authors discuss their findings and analyze their data in comparison with their previous publication.

First, I think that they need to better contextualize their study in the field of population ecology. As stated above, I think that this study does not relate directly to biogeography, and is more related to population ecology. Thus, while the biogeographical bibliography is sound, I feel like more emphasis should be put on the ecology of intertidal ecosystems. A huge amount of work has been produced on such systems that explore many aspects of variations in population densities, including trophic interactions, competition for space, and differential impact of ecological factors. Integrating such questions to the present paper could relevantly improve the discussion and feed the debate on the relative importance of abundance drivers across species range.

Thus, the interpretation (discussion) of the results is very supportive of the ACH, but I think that their results do not support their arguments. Therefore, I would advise authors to tone down their global message.

Below are some other comments on particular points of each section.

Introduction
Bibliographical references are not efficiently displaced within the text, and do not support the reasoning. There is no need to cite ~10 papers in a quote at the end of each paragraph : I would rather read few useful references cited at the end of each idea.

L66 : “ a well replicated study”
Please remain neutral when evocating your own work.

Material and methods
Methods need to be clarified as too few information are available to allow readers to clearly understand the sampling procedure. For example :
- how distant are vertical transects from one another?
- How did you place vertical transects within the bay (randomly, stratified…?)

The study organism is not well presented in the MS. Please :
- Provide a map of its distribution
- Add on the map the location of your sampling site
- Bring information on its ecology (life duration, etc.). Are these animals mobile to some extent?

Results
Due to the very limited dataset, I think that the results are quite weak and do not bring insightful information to the question. The interesting point here would be the comparison of those new results with the one published some years ago. However, I don’t know if it is technically possible due to way field work was conceived…?

Discussion
Authors discuss their result in regard of their previous publication, which is an interesting approach. However, they highlight some changes in the ‘altitude’ where they observe highest local densities, but one cannot assess if this results from a true elevational shift, of if it’s an artefact due to different vertical extents of the two experiments. If authors have access to absolute elevational data of each quadrat (using a reference point common to the two studies) they could strengthen their reasoning and analyze both dataset conjointly.

Reviewer 2 ·

Basic reporting

The article is clear and unambiguous, and professional English used throughout. In addition, the writing style is excellent and the authors provide sufficient field context.

Experimental design

The research question is well defined and the investigation has been performed rigorously. I have just some comments on details needed in the statistical analysis section:

About the (generalised) additive model, I would suggest the authors including the information of how they checked for diagnostics of the analysis. The function gam.check() and plots of the component smooth functions (and residuals) on the linear predictor' scale would be useful here. In this line, how did the author determine what kind of model was the appropriate? I assume that a Gaussian model was used, but the type of data (i.e. density and body size) suggests that maybe other kinds of models would be needed. For instance, the range of density is, theoretically, [0, Inf[, and that of body size would be ]0, Inf[. According to those ranges, Gamma and maybe log-normal distributions would provide a good fit for densities and sizes, respectively (see Bolker 2008 – Ecological Models and Data in R, Princeton University Press, Princeton NJ). Please note that I am not saying that your models are wrong, I am asking for additional information on how the model’ error distribution was selected.

Some other details would be useful to provide; for instance, I would indicate the basis dimension (k) used for the smoothing parameter (elevation). The mgcv::gam function uses a default k = 10 (Wood, S.N. 2006), but there are only seven possible “steps” in the factor “elevation zone”. In addition, the estimated degrees of freedom of each fit would be useful to assess the strength and "curvature" of the smoothed trends.

Line 119: Using a \deltaAICc = 2 as threshold for selecting (or discarding) a given model. Please note that this is a rough rule of thumb (Burnham and Anderson, 2002) to assess the loss of information of a given model (in terms of Kullback-Leibler distance). The larger the delta, the less plausible it is that the fitted model is the best K-L model given the data. The range delta = [0, 2] would indicate that the candidate model (your null model in this case) has substantial level of empirical support; and values [4, 7] denote considerably less support (Burnham and Anderson, 2002; 2004). Indeed, Harrison et al. (2018) indicate that value of delta of 6 would be needed to discard uninformative models. Nevertheless, in this case in particular, the deltas were comparatively high (>10 units), which provides good support for the full models over the null model.

Burnham, K. P. and Anderson, D. R. 2002. Model Selection and Multi-Model Inference -Springer Verlag.

Burnham, K. P. and Anderson, D. R. 2004. Multimodel inference - understanding AIC and BIC in model selection. - Sociol Method Res 33: 261-304

Harrison, X. A., Donaldson, L., Correa-Cano, M. E., Evans, J., Fisher, D. N., Goodwin, C. E., Robinson, B. S., Hodgson, D. J. and Inger, R. 2018. A brief introduction to mixed effects modelling and multi-model inference in ecology. - Peerj 6.

Wood, S. N. 2006. Generalized Additive Models: An Introduction with R. - Taylor & Francis.

Validity of the findings

I believe this point will be fulfilled after the authors provide more details of model validation (please see previous points). This is by no means saying that the results are "invalid", but some more details are needed.

Additional comments

Some specific comments:

Line 42: Habitat conditions —are you referring to abiotic conditions or also to biotic factors? Biotic interactions can influence the abundance of species at local, but also regional spatial scales (e.g. Godsoe et al. 2017).

Line 55: Semibalanus balanoides as an exclusively marine intertidal sessile species. I think this is an important premise of this study, because it allows us to assume that the low intertidal or shallow subtidal represent “sub-optimal” environmental conditions for this species. You may want to provide a reference supporting this statement. Scrosati and Watts (2007) analysed intertidal, but not subtidal, habitats. Buschbaum (2002) may be useful here, albeit the work was conducted on European shores (intertidal and subitidal habitats). (Buschbaum, C. 2002. Predation on barnacles of intertidal and subtidal mussel beds in the Wadden Sea. - Helg Mar Res 56: 37-43.)

Line 78: What do you mean with ecological performance in this context? Competitive ability?

Line 143 – 144: I am not sure that density peaked with such a remarkable accuracy. Actually, the fit shows larger differences between predicted and observed values (residuals) in the middle of the range. I would suggest again checking if the gaussian was the more appropriate model here.

Line 144: OK, the information criteria was useful for selecting the K-S model in the set (actually one model against the null model). However, if all models are poor, AIC will still select the one estimated as the best (Burnham and Anderson, 2002). I would suggest the authors to include in the methods other model validation methods in addition with the explained deviance.

Line 155: Please note that the analyses of Sagarin and Gaines (2002) provided support for the ACH for only two species out of twelve in terms of mean densities, and one out of twelve in terms of maximum density. A similar case is that of Rivadeneira et al. (2010): the ACH was supported in a small proportion of the analysed species.

Line 157 (paragraph): I agree — explaining the differences between two points in time is very difficult.

Line 169-170: I do not understand this idea. Why is the prediction on body size more simplistic than on density?

Line 165 (paragraph): Body size. It is interesting that Rivadeneira et al. (2010) found no statistical fit of latitude with the body size of porcelain crabs, but a slight monotonic increase of the latter towards higher latitudes—which is similar in some degree to your result. Perhaps you may want to mention this in this section.

---

## Round 0.2 · accepted · Accept

The one previous reviewer who was available to examine the revised manuscript and I agree that the changes are satisfactory and that the manuscript is now suitable for publication.

# Reviewer 2 ·

Basic reporting

The authors have made a compelling work including reviewers' suggestions.

Experimental design

No comments

Validity of the findings

The incorporation of model diagnostics allows now to validate of models.

Additional comments

I think the manuscript is ready to be published in PeerJ.